# Comparison of angle-closure detection between automated gonioscopy and anterior-segment optical coherence tomography

Yuki Takagi[1,2]*, Ryo Asano[1,3], Yui Morioka[2], Yukihiro Sakai[2], Sho Yokoyama[1,2], Kei Ichikawa[2], Kazuo Ichikawa[2]

1 Chukyo Hospital, Sanjo 1-chome, Mina-ku, Nagoya, Aichi, Japan, 2 Chukyo Eye Clinic, Sanbonmatsu-cho, Atsuta-ku, Nagoya, Aichi, Japan, 3 Asano Eye Clinic, Iriba 1-chome, Minato-ku, Nagoya, Aichi, Japan

* ytakagi@sanjogroup.jp

## Abstract

### Purpose

To investigate the concordance between angle-closure assessments based on GS-1 gonioscope images and those obtained with anterior-segment optical coherence tomography.

### Study design

Retrospective clinical study.

### Methods

We included 33 patients (53 eyes) who visited Chukyo Eye Clinic during 2020–2024, were suspected of having angle closure, and underwent anterior-segment optical coherence tomography (CASIA2 Advance STAR Analyzer) and GS-1 examinations. The 16-directional images captured with the GS-1 were divided into two halves, creating 32 directions, which were rearranged to correspond with those obtained via anterior-segment optical coherence tomography. Agreement between evaluations was analyzed using Cohen's κ, and the area under the receiver operating characteristic curve was evaluated. Anterior-segment optical coherence tomography images were manually corrected, and eyes with areas classified as "narrow" or "closed" were categorized as angle closure. With the GS-1, two glaucoma specialists independently reviewed the images. Areas in which the posterior trabecular meshwork was obscured in more than half of the image (Scheie classification grades III–IV) were judged indicative of angle closure.

### Results

We included 1,660 directions from 53 eyes in the agreement analysis. The proportion of directions judged as angle closure was 57.0% with anterior-segment optical

**Data availability statement:** All relevant data are within the manuscript and its Supporting Information files.

**Funding:** The author(s) received no specific funding for this work.

**Competing interests:** The authors have declared that no competing interests exist.

coherence tomography and 46.1% with the GS-1. Cohen's κ for inter-test agreement was 0.173 (95% confidence interval: 0.128–0.218), and the area under the receiver operating characteristic curve was 0.588 (95% confidence interval: 0.561–0.615).

## Conclusion

Analyses using anterior-segment optical coherence tomography yielded more frequent classifications of angle closure than evaluations based on GS-1 gonioscopic images.

## Introduction

Gonioscopy, developed in the 1800s [1], is essential for the diagnosis of certain types of glaucoma, particularly angle-closure glaucoma [2,3]. Angle-closure glaucoma is especially important in Asian countries, where it is more common, including in Japan, and less common among Caucasians [4].

Gonioscopy enables angle assessment via color imaging and facilitates static and dynamic examinations, making it useful for evaluating angle structures [5]. However, owing to the complexity of the procedure and the required proficiency, it is not commonly performed in clinical practice [5,6]. Moreover, the evaluation of gonioscopic features reportedly varies depending on examiner expertise [7]. Furthermore, capturing and storing numerous detailed gonioscopic images of the entire angle is difficult. These factors highlight the need for relatively noninvasive devices that can replace or supplement gonioscopy and can be operated by examiners such as orthoptists.

As alternatives to gonioscopy, anterior-segment optical coherence tomography (AS-OCT) and ultrasound biomicroscopy (UBM) can be performed noninvasively and are reportedly useful for evaluating the anterior chamber and iridocorneal angle structures [8–11]. In particular, AS-OCT is highly useful because of its short examination time. However, AS-OCT and UBM cannot assess trabecular meshwork pigmentation or distinguish between organic and functional angle closure. Therefore, these tests have not yet replaced gonioscopy.

The GS-1 Gonioscope (GS-1; NIDEK, Gamagori, Japan) is a device capable of automatically capturing color photographs of the entire angle (16 directions) (Fig 1). Reportedly, it enables rapid examinations with relatively low invasiveness [12–15]. Therefore, it is considered useful for monitoring temporal changes in angle structures, and its application in evaluating peripheral anterior synechiae formation following outflow reconstruction surgery has been reported [16]. As such, it has the potential to complement conventional gonioscopy.

In comparative studies of device performance for angle-closure assessment, angle closure has been more frequently diagnosed than gonioscopy using AS-OCT [17–21]. Gonioscopy has also been compared with EyeCam imaging [22,23], revealing high concordance between them. However, no studies have compared GS-1 with AS-OCT, nor have reports evaluated the diagnostic value of GS-1 images for angle closure. Therefore, in this study, we examined the concordance between

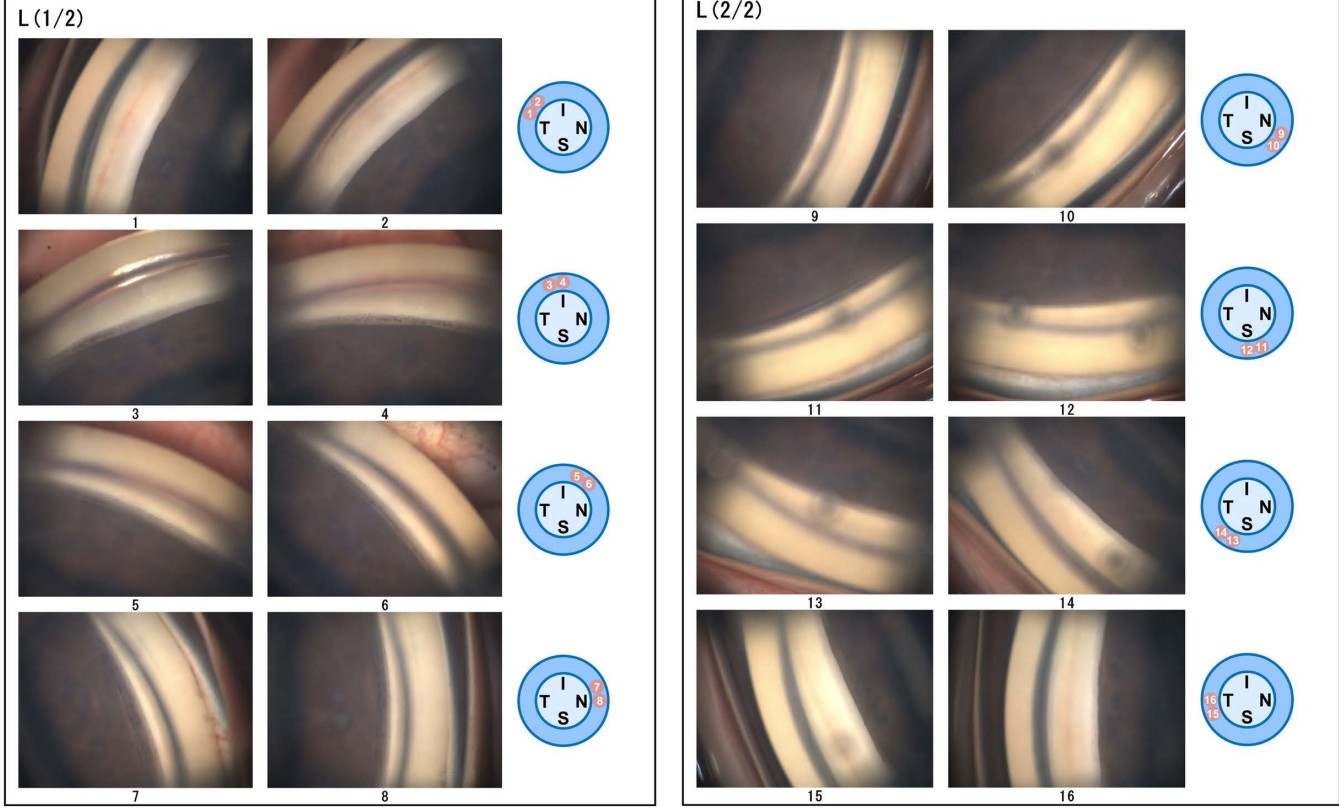

**Fig 1. Gonioscopic Images Captured by Automated Gonioscopy (GS-1).** Gonioscopic photographs of the left (L) eye, with 16 directional angle images captured. Similar to traditional gonioscopy, the displayed image is inverted vertically and horizontally. However, the specific anatomical locations of each angle are indicated next to the corresponding gonioscopic images. I: inferior, N: nasal, S: superior, T: temporal.

angle-closure assessments based on GS-1 and AS-OCT, both of which can be performed by orthoptists. AS-OCT yielded more frequent classifications of angle closure than GS-1.

## Materials and methods

### Patients and study design

Patients who visited Chukyo Eye Clinic from 2020 to 2024, suspected of having angle closure based on a Van Herick grade ≤2 on slit-lamp microscopy, and underwent AS-OCT (CASIA2 Advance; TOMEY, Nagoya, Japan) and GS-1 examinations were included. Exclusion criteria included a history of peripheral iridotomy, intraocular surgeries such as cataract extraction, or ocular diseases other than cataracts.

This study was conducted retrospectively and was reviewed for research purposes on March 3, 2024. It was performed in accordance with the Declaration of Helsinki and approved by the Ethics Committee of Chukyo Eye Clinic (approval number: 20240227075). An opt-out approach was used for informed consent owing to the retrospective nature of the study.

### AS-OCT imaging

In this study, AS-OCT imaging was performed using the CASIA2 Advance device in a dark room by an orthoptist. To assess angle closure, we used the STAR Analyzer program installed on the device. This program automatically classifies

the 32 directions of the anterior chamber angle as open, narrow, or closed based on the angle-opening distance (AOD) at 500 μm anterior to the scleral spur (AOD500) (Fig 2). Cutoff values for the AOD500 in each direction are predefined in the program (ranging from 0.098 to 0.198 mm). If the AOD500 value is equal to or greater than the cutoff, the angle is classified as open. If it is below the cutoff, the angle is classified as narrow; if it is 0 mm, it is classified as closed. Automatic segmentation was first performed for the scleral spur, iris, and other structures. The same examiner then reviewed each image and manually corrected the segmentation before reanalysis of the data. For the final classification, each of the 32-directional results was categorized as "angle closure present" if classified as narrow or closed, and as "angle closure absent" if classified as open.

## Automated gonioscopy imaging

GS-1 imaging was performed by orthoptists in a bright room. Images were captured from 16 directions, divided into halves, and independently reviewed by two glaucoma specialists (R.A. and Y.T.). Areas in which the posterior trabecular meshwork was obscured in more than half of the image (Scheie classification grades III–IV) were judged as indicative of angle closure. In case of disagreements between the two reviewers, a consensus was reached through discussion.

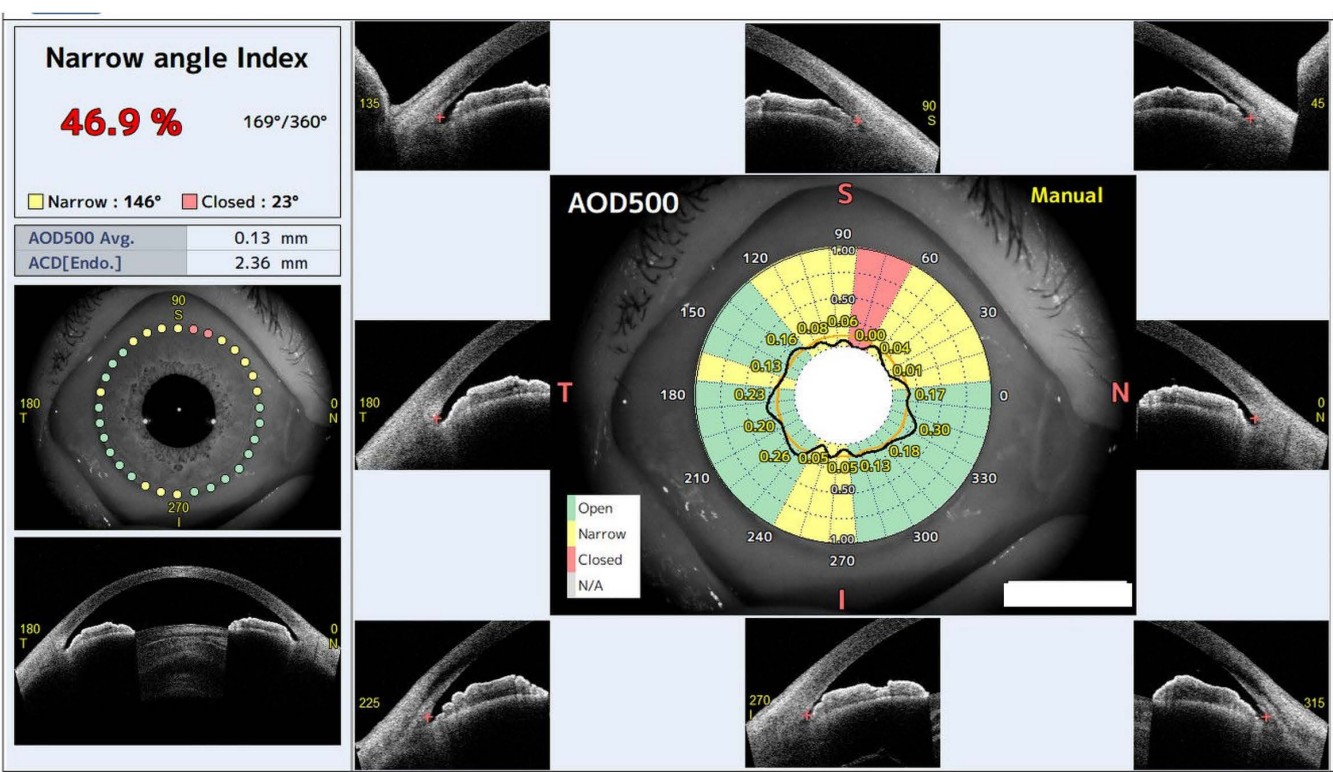

**Fig 2. Example of CASIA2 Advance STAR Analyzer Results.** In the center of the image, the classification results for each of the 32 directions are displayed, with narrow areas shown in yellow, closed areas in red, and open areas in green. In the top left corner, the Narrow Angle Index, a measurement of the degree of angle closure, is displayed. AOD500: angle-opening distance at 500 μm anterior to the scleral spur. I: inferior, N: nasal, S: superior, T: temporal.

## Statistical analysis

The 16-directional images captured via GS-1 were divided into halves, creating 32 directions, which were rearranged to correspond with the 32 directions of the manually corrected AS-OCT results. GS-1 images in which angle structures could not be identified and AS-OCT images in which the scleral spur could not be identified were excluded from the analysis. Cohen's κ was used to evaluate the agreement between the assessments. Additionally, cases in which more than 50% of the images were indicative of angle closure were categorized as angle-closure cases, and the agreement between the manually corrected AS-OCT results and GS-1 results was also evaluated using Cohen's κ.

Furthermore, the images were divided into quadrants (superior, inferior, temporal, and nasal), and the agreement between the methods was assessed for each quadrant using Cohen's κ. Receiver operating characteristic (ROC) analysis was performed, with AS-OCT as the reference standard and GS-1 classification results as the test variables. The area under the ROC curve (AUC) and its 95% confidence interval (CI) were calculated. The agreement of angle closure between the automatic and manually corrected AS-OCT results was also examined for each image and each case.

To evaluate the repeatability of the manual correction process in AS-OCT post-processing, approximately 30% of the total cases were randomly selected for intra-examiner reliability analysis. The same examiner performed a second round of segmentation corrections on a different day under masked conditions (blinded to the first assessment results). Agreement between the two assessments was evaluated using Cohen's κ on a per-image and per-case basis.

All statistical analyses were performed using SPSS (version 29.0; IBM Corp., Armonk, NY, USA), and statistical significance was set at $P < 0.05$.

An a priori power analysis was conducted using G*Power version 3.1.9.7 [24] to determine the required sample size for Cohen's κ analysis. Assuming a medium effect size ($κ = 0.4$), an alpha level of 0.05, and a desired power of 0.80, the minimum required sample size was calculated as 45 eyes.

## Results

A total of 33 patients (53 eyes; 6 males and 27 females; mean ± standard deviation age, 71.3 ± 9.3 years) were included in the study. The analysis covered 1,696 directions across the 53 eyes; however, 8 directions with poor angle visualization upon AS-OCT and 28 directions with unclear images on GS-1 examination were excluded from the agreement analysis. The clinical characteristics of the study population are summarized in Table 1.

**Table 1. Demographic Information of Patients.**

| Characteristics | Total (n = 53) |
|---|---|
| Age, years | 71.3 ± 9.3 |
| Sex, female, number (%) | 42 (79.2%) |
| Number of right eyes (%) | 28 (52.8%) |
| Best-corrected visual acuity (logMAR) | −0.053 ± 0.19 |
| Spherical equivalent (D) | 0.77 ± 2.07 |
| Angle opening distance $_{temporal}$ (mm) | 0.16 ± 0.68 |
| Angle recess area $_{temporal}$ (mm$^2$) | 0.076 ± 0.30 |
| Trabecular iris space area $_{temporal}$ (mm$^2$) | 0.70 ± 0.29 |
| Trabecular iris angle (degree) | 14.44 ± 7.07 |
| Anterior chamber distance (mm) | 2.22 ± 0.18 |
| Lens vault (mm) | 0.76 ± 0.20 |

Values are presented as means ± standard deviations, unless otherwise indicated. logMAR: logarithm of the minimum angle of resolution; D: diopter.

The comparison of the two methods is presented in Table 2. The proportion of images judged as angle closure was 57.0% with AS-OCT and 46.1% with the GS-1. Cohen's κ for inter-test agreement was 0.173 (95% CI: 0.128–0.218). When analyzed by eye, the proportion of eyes judged as angle closure was 60.4% with AS-OCT and 33.9% with GS-1. Cohen's κ was 0.151 (95% CI: −0.076–0.378).

The quadrant-specific comparison results are shown in Table 3. Cohen's κ and the AUC for the superior quadrant were 0.260 (95% CI: 0.168–0.352) and 0.631 (95% CI: 0.578–0.685), respectively; those for the inferior quadrant were 0.203 (95% CI: 0.109–0.297) and 0.605 (95% CI: 0.548–0.661), respectively; those for the nasal quadrant were 0.206 (95% CI: 0.140–0.271) and 0.619 (95% CI: 0.565–0.672), respectively; and those for the temporal quadrant were 0.036 (95% CI: −0.060–0.130) and 0.518 (95% CI: 0.462–0.574), respectively. In the comparison before and after manual correction of AS-OCT measurements, Cohen's κ by image was 0.806 (95% CI: 0.777–0.835), and that by case was 0.763 (95% CI: 0.585–0.941).

In total, 19 eyes were re-evaluated for intra-examiner reliability. The per-image analysis yielded a Cohen's κ of 0.895 (P<0.001), and the per-case analysis demonstrated perfect agreement with a κ of 1.000 (P<0.001). These results indicate high repeatability and consistency of the examiner's manual corrections in this study.

Representative images illustrating discrepancies between GS-1- and AS-OCT-based assessments are provided in Fig 3. In one case, a 73-year-old female patient was judged to have angle closure in 93.8% of directions via AS-OCT but in only 46.8% via GS-1. The inferonasal angle (240°–270°) was deemed closed on AS-OCT; however, the scleral spur was visible in the corresponding GS-1 image, leading to a judgment of an open angle.

## Discussion

In this study, we examined the concordance of angle-closure evaluations between GS-1 and AS-OCT. A higher proportion of AS-OCT images were judged as angle closure compared with GS-1 images. Consequently, in certain cases, AS-OCT indicated notably narrow angles, whereas GS-1 enabled visualization of the trabecular meshwork.

AS-OCT and gonioscopy have been compared in several previous studies [17–21]. Although the AS-OCT devices used in those studies varied, the criteria for angle closure were similar. Angle closure was defined as contact between

**Table 2. Comparison of GS-1 and AS-OCT Assessment of Angle Closure.**

|  | n | Angle closure determined via GS-1 (%) | Angle closure determined via AS-OCT (%) | κ (95% CI) | AUC (95% CI) |
|---|---|---|---|---|---|
| **Per image** | 1,660 | 46.1 | 57.0 | 0.173 (0.128, 0.218) | 0.589 (0.562, 0.617) |
| **Per eye** | 53 | 33.9 | 60.4 | 0.151 (−0.076, 0.378) | 0.584 (0.428, 0.740) |

AS-OCT: anterior-segment optical coherence tomography, AUC: area under the receiver operating characteristic curve, CI: confidence interval, GS-1: GS-1 Gonioscope.

**Table 3. Comparison of GS-1 and AS-OCT Assessment of Angle Closure by Quadrant.**

|  | Angle closure determined via GS-1 (%) | Angle closure determined via AS-OCT (%) | κ (95% CI) | AUC (95% CI) |
|---|---|---|---|---|
| Temporal | 60.0 | 53.75 | 0.036 (−0.060, 0.130) | 0.518 (0.462, 0.574) |
| Nasal | 22.0 | 59.81 | 0.206 (0.140, 0.271) | 0.619 (0.565, 0.672) |
| Superior | 46.06 | 53.7 | 0.260 (0.168, 0.352) | 0.631 (0.578, 0.685) |
| Inferior | 56.35 | 62.35 | 0.203 (0.109, 0.297) | 0.605 (0.548, 0.661) |

AS-OCT: anterior-segment optical coherence tomography, AUC: area under the receiver operating characteristic curve, CI: confidence interval, GS-1: GS-1 Gonioscope.

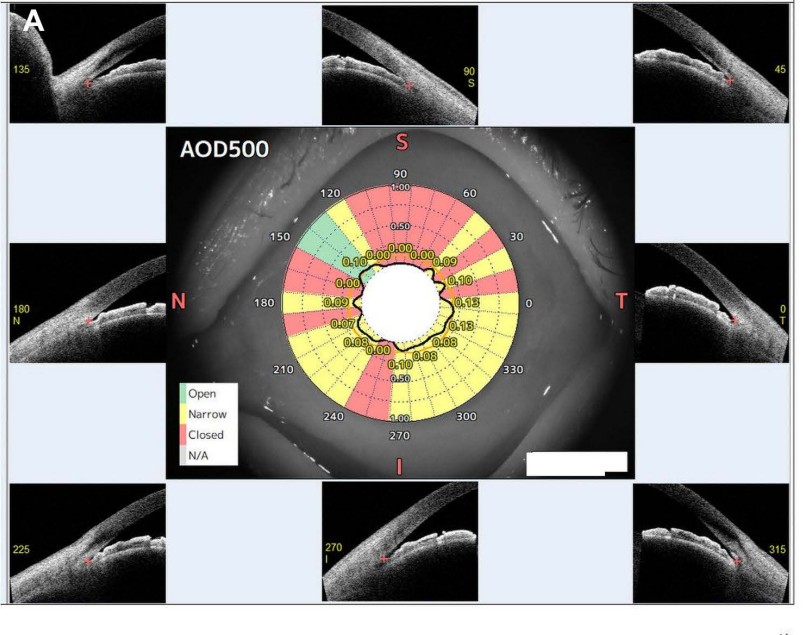

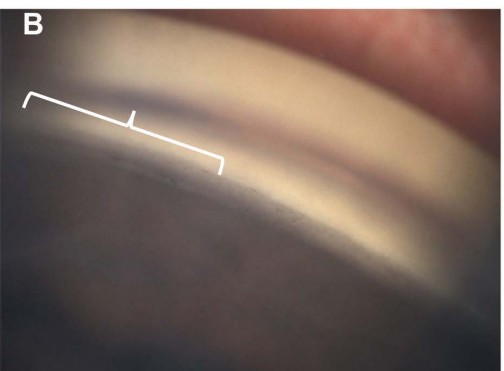

**Fig 3. Example of a Case with Disagreement between AS-OCT and Automated Gonioscopy. A)** Result obtained using the CASIA2 Advance STAR Analyzer, showing "narrow" and "closed" angles in 93.8% of directions. The superior quadrant appears wide, whereas the inferior quadrant exhibits closure in the 240°–270° range. **B)** Corresponding GS-1 image of the inferior quadrant (240°–270°). In the left half of the image, the trabecular meshwork and scleral spur are visible (white brace). I: inferior, N: nasal, S: superior, T: temporal.

the scleral spur and the iris with AS-OCT, whereas it was defined as Scheie classification grade III–IV with gonioscopy. In all of those reports, angle closure was more frequently judged with AS-OCT than with gonioscopy [17–21], with κ-values reported as moderate [20].

In this study, the criteria for angle closure differed from those in the abovementioned reports [17–21], and automated rather than manual gonioscopy was used. However, AS-OCT still resulted in a higher proportion of angle-closure judgments. Conversely, Cohen's κ was lower than the moderate agreement reported in previous studies, a notable difference.

Several factors may explain this discrepancy. One potential factor is the use of GS-1 instead of manual gonioscopy. During GS-1 imaging, the device emits light, which enters the pupil and may induce constriction. Angles reportedly appear wider under bright conditions than under dark conditions [25,26]. Therefore, although GS-1 examinations performed under bright conditions allow 360° color imaging that is useful for documenting features such as pigmentation, they may not be suitable for screening or diagnosing angle closure. A similar influence has been suggested in a previous report using devices such as the EyeCam [22]. However, that report demonstrated moderate agreement, differing from the results of this study.

Another possible explanation is the variation in the criteria used to determine angle closure with AS-OCT between previous reports and this study. In prior studies [17–21], angle closure was defined as contact between the scleral spur and the iris. In contrast, this study used the CASIA2 Advance scanner setting of AOD500 as the threshold. Further research is required to standardize the criteria for angle closure across methods.

For gonioscopy, manual and automated, the criteria in previous reports and in this study were based on the visibility of the posterior trabecular meshwork. Anatomically, the visibility of the posterior trabecular meshwork and scleral spur–iris contact are considered relatively comparable criteria. In contrast, AOD500 and posterior trabecular meshwork visibility are

structurally different measures. This discrepancy likely accounts for the lower concordance observed in this study compared with previous reports [20,22].

In the quadrant-specific comparison, the superior quadrant had the highest κ-value. A previous report using UBM indicated that the angle width in the superior quadrant does not significantly differ between dark and bright conditions [26], suggesting that the superior quadrant is the least influenced by illumination among all the quadrants, as also implied by the results of this study.

In the inferior quadrant, AS-OCT yielded more frequent judgments of angle closure than did gonioscopic evaluations with the GS-1. The abovementioned report [26] indicated that the inferior quadrant has the narrowest angle in dark conditions. Thus, the inferior quadrant may be most prone to narrowing in dark environments, and since AS-OCT in this study was performed under dark conditions, the results align with those of the previous report.

That prior study [26] also revealed that the difference in angle width between dark and bright conditions was greatest in the nasal quadrant. Consistently, in this study, the nasal quadrant exhibited the largest disparity in angle-closure assessments between GS-1 and AS-OCT.

Furthermore, the previous study [26] reported that the angle in the temporal quadrant was generally wider than in the other quadrants under bright and dark conditions, with smaller differences between lighting conditions compared with the inferior and nasal quadrants. However, in this study, GS-1 identified angle closure more frequently in the temporal quadrant than in the other quadrants, suggesting a deviation from previous findings. A potential explanation for this discrepancy is that GS-1 imaging involves a degree of ocular compression. However, the temporal quadrant may be the least affected by such compression during GS-1 imaging among all four quadrants. Therefore, larger studies incorporating comparisons with manual gonioscopy are warranted to determine whether these tendencies are consistent.

In this study, AS-OCT evaluations were performed using the CASIA2 Advance STAR Analyzer, with manual segmentation corrections and AOD500 employed to assess angle closure. The segmentation corrections involved not only adjusting the position of the scleral spur but also revising the iris segmentation, which is essential for AOD500. In most cases, manual corrections of the iris segmentation were necessary. The primary reason was that, in cases with narrow angles, the segmentation often did not extend to the iris root. Consequently, much of the manual work involved extending the segmentation to the iris root. Despite this, the agreement of angle closure between the automatic analysis of the CASIA2 Advance STAR Analyzer and the manually corrected results was high. This likely reflects the fact that cases requiring segmentation corrections predominantly consisted of those with narrow angles. Therefore, the high agreement can be attributed to the narrower angles that necessitated corrections. Given this observation, the automatic analysis of the CASIA2 Advance STAR Analyzer may still be useful for screening angle closure. However, we did not analyze the extent of the required corrections in this study, nor the proportion of cases requiring correction based on the presence or severity of angle closure. These aspects warrant further investigation.

This study has some limitations. The number of cases was limited, and larger investigations are necessary. Comparisons with manual gonioscopy were not performed, and uncertainty remains regarding the extent to which the GS-1 captures the angle in a more open configuration compared with manual gonioscopy. In addition, because AS-OCT was used as a relative reference standard, verification bias is possible. Therefore, future studies are needed that include three-way comparisons among GS-1, manual gonioscopy, and AS-OCT. The concordance between angle-closure determination using the CASIA2 Advance STAR Analyzer, which employs AOD500 as a criterion, and manual gonioscopy features remains unclear and warrants future investigation. In this study, angles classified as either "narrow" or "closed" via AS-OCT were collectively considered closed angles in comparison with GS-1 results. This approach was taken to ensure consistency and fairness in the comparison, given that GS-1 provides only qualitative, static images under bright conditions without dynamic or indentation assessment, complicating the discernment of true appositional closure and a merely narrow angle. Although AS-OCT measurements are more sensitive and quantitative, no standardized grading system for GS-1 results has been established, and subjective judgments can easily influence interpretation. Therefore, we adopted

a binary classification of "open" versus "non-open" in this study. However, more detailed comparisons, in which "narrow" and "closed" categories are separated, should be considered in future research. Another limitation of this study is that we did not formally evaluate intra- and inter-observer agreement in the interpretation of GS-1 images. Although all images were independently assessed by two glaucoma specialists and final decisions were made through consensus in cases of disagreement, we did not calculate numerical agreement rates or have the same examiner perform repeat evaluations. In future studies, the incorporation of reliability analyses, using statistics such as Cohen's κ or the intraclass correlation coefficient, may enhance the validity and reproducibility of the evaluation methods.

In conclusion, the analysis results for angle closure via the CASIA2 Advance STAR Analyzer yielded more frequent judgments of angle closure than the evaluations based on gonioscopic images captured with the GS-1. As AS-OCT can be performed in a dark room, it may be more useful than the GS-1 for the diagnosis of angle closure. However, for evaluating findings such as angle pigmentation, the GS-1's capability to capture and archive 360° color images of the entire angle may be useful.

## Supporting information

**S1 File. Clinical data of all included patients.**
(XLSX)

## Acknowledgments

The authors would like to thank Editage for their English language review.

## Author contributions

**Data curation:** Yui Morioka, Yukihiro Sakai.

**Formal analysis:** Yuki Takagi.

**Supervision:** Kazuo Ichikawa.

**Writing – original draft:** Yuki Takagi.

**Writing – review & editing:** Ryo Asano, Sho Yokoyama, Kei Ichikawa.

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
