## [Decision Letter · Decision Letter 0]

20 Jun 2025

Dear Dr. Takagi,

We look forward to receiving your revised manuscript.

Kind regards,

Jiro Kogo

Academic Editor

PLOS ONE

**Journal Requirements:**

1. When submitting your revision, we need you to address these additional requirements. Please ensure that your manuscript meets PLOS ONE's style requirements, including those for file naming. The PLOS ONE style templates can be found at https://journals.plos.org/plosone/s/file?id=wjVg/PLOSOne_formatting_sample_main_body.pdf and https://journals.plos.org/plosone/s/file?id=ba62/PLOSOne_formatting_sample_title_authors_affiliations.pdf 2. Please include captions for your Supporting Information files at the end of your manuscript, and update any in-text citations to match accordingly. Please see our Supporting Information guidelines for more information: http://journals.plos.org/plosone/s/supporting-information.

Reviewers' comments:

Reviewer's Responses to Questions

**Comments to the Author**

1. Is the manuscript technically sound, and do the data support the conclusions?

Reviewer #1: Partly

Reviewer #2: Yes

Reviewer #3: Yes

2. Has the statistical analysis been performed appropriately and rigorously?

Reviewer #1: No

Reviewer #2: Yes

Reviewer #3: Yes

3. Have the authors made all data underlying the findings in their manuscript fully available?

Reviewer #1: Yes

Reviewer #2: Yes

Reviewer #3: Yes

4. Is the manuscript presented in an intelligible fashion and written in standard English?

Reviewer #1: Yes

Reviewer #2: Yes

Reviewer #3: Yes

**Reviewer #1:**  I appreciate the authors’ effort to evaluate concordance between AS-OCT and GS-1 gonioscopic photography for angle-closure assessment. However, I have identified several critical methodological limitations that, in the present design and with available data, cannot be adequately addressed:

1. Sample size and power calculation

You acknowledge in the Limitations section that the study’s sample size is small. However, no formal sample size or power calculation is reported. Without evidence that your study was sufficiently powered to detect clinically meaningful differences in concordance, the validity of your findings remains uncertain.

2. Choice of comparison standard

Gonioscopy is rightly considered the gold standard for assessing angle closure. Yet your study compares AS-OCT and GS-1 images to one another, rather than directly to gonioscopic grading. Logically, to evaluate whether GS-1 photography can serve as a less invasive alternative, its results should be compared against gonioscopic examination, not AS-OCT.

3. Inconsistent lighting conditions

AS-OCT examinations were performed under dark-room conditions, whereas GS-1 photographs were acquired in ambient light. These differing illumination environments can substantially affect angle appearance, rendering direct comparison inappropriate.

4. Intra-examiner reliability during manual correction

During the AS-OCT post-processing stage, an examiner manually corrected the automated segmentation. It is unclear whether intra-examiner consistency of these corrections was assessed, and no reliability statistics are provided.

5. Undefined reference standard in ROC analysis

In your ROC curve analysis, you do not specify which modality serves as the reference (gold standard). Without this clarification, the results cannot be interpreted.

Given these fundamental methodological flaws, we believe the manuscript is not suitable for further consideration in its present form. We encourage you to address these issues—particularly defining an appropriate gold standard, standardizing imaging conditions, and performing rigorous reliability and power analyses—before resubmitting, either to this journal or elsewhere.

**Reviewer #2: ** The paper conducted an agreement analysis of angle-closure assessments between GS-1 and AS-OCT. While the experiments and analysis are clearly presented, a few issues remain in the explanations, such as unclear comparison restrictions and limitations in the dataset. The following concerns are raised:

1.Definition of angle closure in AS-OCT and GS-1: In AS-OCT, angles classified as “narrow” or “closed” are considered as angle closure, while GS-1 defines angle closure as the obscuration of the posterior trabecular meshwork for more than half of the eye. Could you clarify how the terms “narrow” or “closed” are specifically defined in AS-OCT?

2.Comparison between AS-OCT slices and GS-1 areas: The comparison between AS-OCT slices and GS-1 areas using directional pairs is presented. However, both the Cohen’s Kappa and AUC are relatively low. When using AOD 500 as a threshold, why is a narrow angle also considered as angle closure, given that AS-OCT was originally found to have higher sensitivity for assessing angle closure severity?

3.Lighting conditions and the comparison with dark-room AS-OCT: As mentioned in the discussion, GS-1 sheds light during the examination, which could be considered a light-examination condition. It is commonly agreed that the anterior angle appears narrower under dark-room conditions. Given this assumption, is the comparison with dark-room AS-OCT results truly valid? Could this lighting difference be a factor influencing the disagreement between the two measurements?

4. Limitations in dataset: Although the statistical analysis has been conducted and the sample size exceeds the minimum required, the number of participants is relatively small, and the distribution of participants is unclear.

**Reviewer #3:**  Thank you for submitting this interesting and clinically relevant study comparing automated gonioscopy (GS-1) with AS-OCT for angle-closure detection. The study design is appropriate, and the topic is important for glaucoma diagnostics.

However, I recommend addressing the following key points before the manuscript is suitable for publication:

Low agreement between methods

• The observed inter-modality agreement is minimal (κ ≈ 0.17). Please provide a more critical discussion of the clinical implications of this low concordance. Should clinicians rely more on one modality?

Inclusion/exclusion criteria and clinical parameters

• Please clarify whether patients were included consecutively or selectively, and provide more detail on patient characteristics such as lens status (phakic/pseudophakic), presence of other eye diseases (e.g., plateau iris), and axial length.

Observer variability

• The manuscript would benefit from reporting intra- and interobserver variability, and from explaining whether repeat assessments were performed or whether consensus procedures were used to resolve discrepancies.

Reproducibility

• It's unclear if repeat measurements or quality control thresholds were applied during imaging or analysis.

Minor points

• Address minor formatting issues (e.g., font size inconsistencies between lines 53–60; spacing issues in Table 1).

• Typo in line 215: “mat” → “may”.

Recommendation:

The manuscript addresses an important clinical topic, but revision is needed to improve methodological clarity and the interpretation of findings, particularly regarding the limitations of GS-1 and the impact of environmental and definitional differences.

**Do you want your identity to be public for this peer review?** For information about this choice, including consent withdrawal, please see our Privacy Policy

Reviewer #1: No

Reviewer #2: No

Reviewer #3: No

---

## [Author Response · Author response to Decision Letter 1]

14 Jul 2025

July 11, 2025

Dr. Jiro Kogo

Academic Editor

PLOS ONE

Revised manuscript: PONE-D-25-17859

Title: “Comparison of angle-closure detection between automated gonioscopy and anterior-segment optical coherence tomography”

Dear Dr. Kogo:

We are pleased to correspond with you again regarding the revision of our aforementioned manuscript, which has benefited from a review that has contributed to improving its academic value.

We are grateful to the reviewers and to you for the thorough review of our revised manuscript. We also appreciate the insightful comments provided by the reviewers, which have significantly helped to improve the quality of our manuscript.

We have revised the manuscript in accordance with both the specific and general requests, with the modified text highlighted in yellow in the revised manuscript. Below, please find our point-by-point responses to the reviewers’ comments and concerns. Our responses are provided in blue text. 

RESPONSE TO REVIEWER’S COMMENTS

REVIEWER EVALUATION

We would like to thank the reviewers for their time and efforts in reviewing our manuscript and for providing comments that have considerably helped us to improve our manuscript.

Reviewer #1: I appreciate the authors’ effort to evaluate concordance between AS-OCT and GS-1 gonioscopic photography for angle-closure assessment. However, I have identified several critical methodological limitations that, in the present design and with available data, cannot be adequately addressed:

1. Sample size and power calculation

You acknowledge in the Limitations section that the study’s sample size is small. However, no formal sample size or power calculation is reported. Without evidence that your study was sufficiently powered to detect clinically meaningful differences in concordance, the validity of your findings remains uncertain.

Thank you for your valuable comment. We agree that conducting an a priori power analysis and determining the required sample size are essential aspects in clinical research. In this study, we performed an a priori power analysis based on Cohen’s kappa by using G*Power version 3.1.9.7 (Faul et al., 2007) to calculate the necessary sample size. Specifically, assuming a medium effect size (κ = 0.4), an alpha level of 0.05, and a desired power of 0.80, the minimum required sample size was calculated to be 45 eyes. This information has been clearly stated in the Methods section, as follows:

“An a priori power analysis was conducted using GPower version 3.1.9.7 [24] to determine the required sample size for Cohen’s κ analysis. Assuming a medium effect size (κ = 0.4), an alpha level of 0.05, and a desired power of 0.80, the minimum required sample size was calculated as 45 eyes.” (Page 10, lines 158-161)

24. Faul F, Erdfelder E, Lang AG, Buchner A. G *Power 3: a flexible statistical power analysis program for the social, behavioral, and biomedical sciences. Behav Res Meth 2007;39:175-91. doi: 10.3758/bf03193146.

2. Choice of comparison standard

Gonioscopy is rightly considered the gold standard for assessing angle closure. Yet your study compares AS-OCT and GS-1 images to one another, rather than directly to gonioscopic grading. Logically, to evaluate whether GS-1 photography can serve as a less invasive alternative, its results should be compared against gonioscopic examination, not AS-OCT.

Thank you for your comment. We fully understand that gonioscopy is the gold standard to diagnose angle closure, and that a comparison of GS-1 results with gonioscopic examination is essential to validate the accuracy of the GS-1. However, in clinical practice, reports have indicated that gonioscopy is often underutilized owing to time and technical constraints. Therefore, alternative or complementary methods that can be performed by examiners other than physicians, such as orthoptists, are needed to screen for angle closure.

AS-OCT has already been demonstrated in several previous studies to be useful for the diagnosis of angle closure. Moreover, AS-OCT can be performed by orthoptists, and with practical clinical implementation in mind, we prioritized a comparison between the GS-1 and AS-OCT in this study.

We also recognize the importance of directly comparing the GS-1 with manual gonioscopy, and we have clearly stated this limitation in the Discussion section, as follows:

“Additionally, comparisons with manual gonioscopy were not performed, leaving uncertainty about the extent to which the GS-1 captures images of the angle in a more open state compared with manual gonioscopy. Therefore, future studies are needed for comparisons between the GS-1 and manual gonioscopy.” (Page 19, lines 300-303)

I also added a statement in the Introduction to emphasize the need for alternative or complementary examinations to gonioscopy that can be performed by examiners such as orthoptists, as follows:

“Gonioscopy allows for angle assessment via color imaging and facilitates both static and dynamic examinations, rendering it useful for the understanding of angle structures [5]. However, owing to the complexity of the procedure and the need for proficiency, it is not always performed in practice [5,6]. Moreover, the evaluation of gonioscopic features reportedly varies depending on the degree of proficiency [7]. Furthermore, capturing and storing numerous detailed gonioscopic images of the entire angle is difficult. These factors reveal a need for relatively non-invasive devices that can replace or supplement gonioscopes and can be operated by examiners such as orthoptists.” (Page 4, lines 46-53)

3. Inconsistent lighting conditions

AS-OCT examinations were performed under dark-room conditions, whereas GS-1 photographs were acquired in ambient light. These differing illumination environments can substantially affect angle appearance, rendering direct comparison inappropriate.

Thank you for your insightful comments. We consider this a very important point. In principle, the standard approach for the diagnosis of angle closure is to perform examinations under dark-room conditions, as the anterior chamber angle reportedly appears wider under bright conditions. Therefore, GS-1 examinations should ideally also be performed in a dark room. However, as the GS-1 device itself emits light during imaging, we judged that the room conditions would not make a difference. Consequently, we performed the GS-1 examinations under normal, light conditions. We acknowledge that this might have significantly influenced the results of our study, and we have described this point in the Discussion section, as follows:

“During GS-1 imaging, the device emits light, causing light to enter the participant’s pupil and potentially induce pupil constriction. Angles reportedly appear wider under bright conditions than they do under dark conditions [25,26]. Therefore, GS-1-based examinations, which are performed under bright conditions, may not be useful for screening or diagnosis of angle closure.” (Page 16, lines 232-236)

As mentioned earlier, the objective of this study was to evaluate the diagnostic capability of GS-1 gonioscopy to detect angle closure, specifically as a modality that can be performed by orthoptists. Similarly, AS-OCT can be operated by orthoptists and its utility in the diagnosis of angle closure has already been reported. Therefore, considering the feasibility of examinations performed by orthoptists, we prioritized a comparison between GS-1 gonioscopy and AS-OCT in this study; clinical implementation was our foremost concern.

4. Intra-examiner reliability during manual correction

During the AS-OCT post-processing stage, an examiner manually corrected the automated segmentation. It is unclear whether intra-examiner consistency of these corrections was assessed, and no reliability statistics are provided.

Thank you for your helpful comments. We agree that intra-examiner reproducibility of AS-OCT is an important factor to ensure its reliability. In this study, approximately 30% of the cases (19 eyes) were randomly selected to assess intra-examiner reproducibility of AS-OCT evaluations. The analysis demonstrated high agreement, with a per-image Cohen’s kappa of 0.895 (P<0.001) and a per-case kappa of 1.000 (P<0.001), indicating excellent intra-examiner consistency in this study.

We have added the following sentences to the Methods and Results sections:

“To evaluate the repeatability of the manual correction process in AS-OCT post-processing, approximately 30% of the total cases were randomly selected for intra-examiner reliability analysis. The same examiner performed a second round of segmentation corrections on a different day under masked conditions (the examiner was blinded to the first assessment results). Agreement between the two assessments was evaluated using Cohen’s κ statistics on both a per-image and per-case basis.” (Pages 9-10, lines 149-154)

“In total, 19 eyes were re-evaluated for intra-examiner reliability. The per-image analysis yielded a Cohen’s κ of 0.895 (P<0.001), and the per-case analysis revealed perfect agreement with a kappa of 1.000 (P<0.001). These results indicate high repeatability and consistency of the examiner’s manual corrections in this study.” (Page 14, lines 193-196)

5. Undefined reference standard in ROC analysis

In your ROC curve analysis, you do not specify which modality serves as the reference (gold standard). Without this clarification, the results cannot be interpreted.

Given these fundamental methodological flaws, we believe the manuscript is not suitable for further consideration in its present form. We encourage you to address these issues—particularly defining an appropriate gold standard, standardizing imaging conditions, and performing rigorous reliability and power analyses—before resubmitting, either to this journal or elsewhere.

Thank you for your comment. We apologize for not clearly stating that AS-OCT was used as the reference standard in the ROC analysis to evaluate concordance with GS-1 gonioscopy. In response to your suggestion, we have now clarified this fact in the Methods section.

“Receiver operating characteristic (ROC) analysis was performed, using AS-OCT as the reference standard and the GS-1 classification results as the test variables. The area under the ROC curve (AUC) and its 95% confidence interval were calculated.” (Page 9, lines 143-145)

Furthermore, upon recalculating and reviewing the analyses, we discovered that the ROC curves had been calculated using GS-1 gonioscopy as the reference standard instead of AS-OCT in some cases. We re-confirmed that AS-OCT is the correct reference standard, and we have repeated the ROC analyses. The corrected results have been updated in Tables 2 and 3 and the related text (page 12, lines 183-187).

Table 2. Comparison of GS-1 and AS-OCT Assessment of Angle Closure.

n Angle closure determined via GS-1 (%) Angle closure determined via AS-OCT (%) κ (95% CI) AUC (95%CI)

Per image 1660 46.1 57.0 0.173 (0.128, 0.218) 0.589 (0.562, 0.617)

Per eye 53 33.9 60.4 0.151 (-0.076, 0.378) 0.584 (0.428, 0.740)

AS-OCT: anterior-segment optical coherence tomography, AUC: area under the receiver operating characteristic curve, CI: confidence interval, GS-1: GS-1 Gonioscope.

Table 3. Comparison of GS-1 and AS-OCT Assessment of Angle Closure by Quadrant.

Angle closure determined via GS-1 (%) Angle closure determined via AS-OCT (%) κ (95% CI) AUC (95% CI)

Temporal 60.0 53.75 0.036 (-0.060, 0.130) 0.518 (0.462, 0.574)

Nasal 22.0 59.81 0.206 (0.140, 0.271) 0.619 (0.565, 0.672)

Superior 46.06 53.7 0.260 (0.168, 0.352) 0.631 (0.578, 0.685)

Inferior 56.35 62.35 0.203 (0.109, 0.297) 0.605 (0.548, 0.661)

AS-OCT: anterior-segment optical coherence tomography, AUC: area under the receiver operating characteristic curve, CI: confidence interval, GS-1: GS-1 Gonioscope.

Reviewer #2: The paper conducted an agreement analysis of angle-closure assessments between GS-1 and AS-OCT. While the experiments and analysis are clearly presented, a few issues remain in the explanations, such as unclear comparison restrictions and limitations in the dataset. The following concerns are raised:

1.Definition of angle closure in AS-OCT and GS-1: In AS-OCT, angles classified as “narrow” or “closed” are considered as angle closure, while GS-1 defines angle closure as the obscuration of the posterior trabecular meshwork for more than half of the eye. Could you clarify how the terms “narrow” or “closed” are specifically defined in AS-OCT?

Thank you for your insightful comments. The criteria for angle closure are indeed a critical aspect of this study. Although the criteria to determine angle closure upon AS-OCT were described in the Methods section, the specific numerical thresholds were not included. We have now added these detailed values.

“To assess angle closure, we used the STAR Analyzer program installed on the device. This program automatically classifies the 32 directions of the anterior chamber angle as open, narrow, or closed based on the angle-opening distance (AOD) at 500 μm anterior to the scleral spur (AOD500) (Fig 2). Cutoff values for the AOD500 in each direction are predefined in the program (ranging from 0.087 to 0.198 mm). If the AOD500 value is equal to or greater than the cutoff, the angle is classified as open. If it is below the cutoff, the angle is classified as narrow, and if it is 0 mm, it is classified as closed.” (Page 7, lines 103-110)

2.Comparison between AS-OCT slices and GS-1 areas: The comparison between AS-OCT slices and GS-1 areas using directional pairs is presented. However, both the Cohen’s Kappa and AUC are relatively low. When using AOD 500 as a threshold, why is a narrow angle also considered as angle closure, given that AS-OCT was originally found to have higher sensitivity for assessing angle closure severity?

Thank you for your comment. In this study, the definition of angle closure via GS-1 gonioscopy, based on previous reports, was the inability to observe the posterior trabecular meshwork. However, this definition relies on qualitative assessment, and since indentation gonioscopy cannot be performed with the GS-1 device, discerning whether the angle is simply narrow or actually appositional (closed) from the images alone is challenging.

Given this background, we determined that GS-1 gonioscopy can only provide a binary classification into “open” and “non-open” (including both narrow and closed angles). Therefore, in this study, we also combined “narrow” and “closed” classifications made via AS-OCT as “angle closure present” to maintain consistency and fairness in the comparison.

However, as AS-OCT allows for more sensitive and quantitative evaluation, we recognize that if standardized evaluation methods for GS-1 become established in the future, separately analyzing “narrow” and “closed” angles would be meaning ful. We have added this point as a supplementary note in the Discussion section, as follows:

“In this study, angles classified as either “narrow” or “closed” via AS-OCT were collectively considered as closed angles in comparison with GS-1 results. This approach was taken to ensure consistency and fairness in the comparison, given that GS-1 provides only qualitative, static images under bright conditions without dynamic or indentation assessment, complicating the discernment of true appositional closure and a merely narrow angle. Although AS-OCT measurements are more sensitive and quantitative, no standardized grading system for GS-1 results has been established, and subjective judgments can easily influence the interpretation. Therefore, we adopted a binary classification of “open” versus “non-open” in this study. However, more detailed comparisons, in which “narrow” and “closed” categories are separated should be considered in future research.” (Page 19-20, lines 306-316)

3.Lighting conditions and the comparison with dark-room AS-OCT: As mentioned in the discussion, GS-1 sheds light during the examination, which could be considered a light-examination condition. It is commonly agreed that the anterior angle appears narrower under dark-room conditions. Given this assumption, is the comparison with dark-room AS-OCT results truly valid? Could this lighting difference be a

---

## [Decision Letter · Decision Letter 1]

26 Aug 2025

Dear Dr. Takagi,

We look forward to receiving your revised manuscript.

Kind regards,

Jiro Kogo

Academic Editor

PLOS ONE

Journal Requirements:

Reviewers' comments:

Reviewer's Responses to Questions

**Comments to the Author**

Reviewer #2: All comments have been addressed

2. Is the manuscript technically sound, and do the data support the conclusions?

Reviewer #2: Yes

3. Has the statistical analysis been performed appropriately and rigorously?

Reviewer #2: Yes

4. Have the authors made all data underlying the findings in their manuscript fully available?

Reviewer #2: Yes

5. Is the manuscript presented in an intelligible fashion and written in standard English?

Reviewer #2: Yes

Reviewer #2: We appreciate the authors’ thorough responses to the previous reviews. However, a few minor concerns remain, and we would appreciate further clarification within the manuscript.

1. In the conclusion section, the authors state that AS-OCT may be more useful than the GS-1 because it can be performed in a dark room. However, traditional gonioscopy is also performed under dark-room conditions, and the GS-1 is introduced as “having the potential to complement conventional gonioscopy”. Could the authors further clarify the rational and intended clinical role of the GS-1 in this context aside from being non-contact (since it was not mentioned after introduction section)? Specifically, what is the motivation for selecting the GS-1 as the comparison method against AS-OCT, rather than manual gonioscopy, given that both AS-OCT and GS-1 differ in imaging principles and lighting conditions?

2. Although being frequently questioned in recent years, manual gonioscopy is still considered as a golden standard for angle closure examination, please justify how AS-OCT results in this case replace manual gonioscopy and be considered as a relative ‘true value’ in the current comparison.

3. The manuscript is generally well-written, though minor grammatical refinements could help with better understanding.

**Do you want your identity to be public for this peer review?** For information about this choice, including consent withdrawal, please see our Privacy Policy

Reviewer #2: No

---

## [Author Response · Author response to Decision Letter 2]

1 Sep 2025

September 1, 2025

Dr. Jiro Kogo

Academic Editor

PLOS One

Revised manuscript: PONE-D-25-17859R1

Title: “Comparison of angle-closure detection between automated gonioscopy and anterior-segment optical coherence tomography”

Dear Dr. Jiro Kogo:

It is a pleasure to correspond with you again regarding the revision of our aforementioned manuscript, which received a favorable review.

We are grateful to the reviewers and the editor for their thorough evaluation of our revised manuscript. We also appreciate the insightful comments provided by the reviewer, which have greatly improved the quality of the work.

We have revised the manuscript in accordance with the specific and general requests, with the modified text highlighted in yellow in the revised manuscript. Below, please find our point-by-point responses to the reviewers’ comments and concerns. Our responses are provided in blue text. 

RESPONSE TO REVIEWER’S COMMENTS

REVIEWER EVALUATION

We would like to thank the reviewers for their time and effort in reviewing our manuscript and for providing comments that have considerably helped us improve it.

Reviewer #2: We appreciate the authors’ thorough responses to the previous reviews. However, a few minor concerns remain, and we would appreciate further clarification within the manuscript.

1. In the conclusion section, the authors state that AS-OCT may be more useful than the GS-1 because it can be performed in a dark room. However, traditional gonioscopy is also performed under dark-room conditions, and the GS-1 is introduced as “having the potential to complement conventional gonioscopy”. Could the authors further clarify the rational and intended clinical role of the GS-1 in this context aside from being non-contact (since it was not mentioned after introduction section)? Specifically, what is the motivation for selecting the GS-1 as the comparison method against AS-OCT, rather than manual gonioscopy, given that both AS-OCT and GS-1 differ in imaging principles and lighting conditions?

Thank you for your valuable comments. We fully understand that gonioscopy is the gold standard for diagnosing angle closure and that comparing GS-1 findings with gonioscopic examination is essential to validate the accuracy of the GS-1. However, in real-world clinical practice, reports indicate that gonioscopy is often underutilized because of time and technical constraints. Therefore, there is a need for alternative or complementary methods that can be performed by examiners other than physicians, such as orthoptists, to screen for angle closure. The primary aim of the present study was to compare AS-OCT and GS-1, both of which can be performed by orthoptists. The need for equipment that can be operated by orthoptists as a potential alternative to manual gonioscopy is described in the Introduction as follows:

“Gonioscopy enables angle assessment via color imaging and facilitates static and dynamic examinations, making it useful for evaluating angle structures [5]. However, owing to the complexity of the procedure and the required proficiency, it is not always performed in clinical practice [5,6]. Moreover, the evaluation of gonioscopic features reportedly varies depending on examiner expertise [7]. Furthermore, capturing and storing numerous detailed gonioscopic images of the entire angle is difficult. These factors highlight the need for relatively noninvasive devices that can replace or supplement gonioscopy and can be operated by examiners such as orthoptists.” (page 4, lines 47-54)

Furthermore, to clarify the study objective, we have revised and expanded the Introduction with the following sentence:

“Therefore, in this study, we examined the concordance between angle-closure assessments based on GS-1 and AS-OCT, both of which can be performed by orthoptists.” (page 6, lines 80-82)

Regarding the clinical role of the GS-1, based on our findings, we consider it unsuitable for screening for angle closure. However, because it captures 360° color images of the entire angle, we believe it is useful for evaluating features such as pigmentation. Accordingly, we have revised and expanded the Discussion section as follows:

“Therefore, although GS-1 examinations performed under bright conditions allow 360° color imaging that is useful for documenting features such as pigmentation, they may not be suitable for screening or diagnosing angle closure.” (page 16, lines 231-233)

“However, for evaluating findings such as angle pigmentation, the GS-1’s capability to capture and archive 360° color images of the entire angle may be useful.” (page 20, line 324 to page 21, line 326)

Finally, we apologize if our wording caused confusion; the GS-1 is not a non-contact device—it is a contact device.

2. Although being frequently questioned in recent years, manual gonioscopy is still considered as a golden standard for angle closure examination, please justify how AS-OCT results in this case replace manual gonioscopy and be considered as a relative ‘true value’ in the current comparison.

Thank you for your comments. As noted above, we likewise consider manual gonioscopy to be the current gold standard. However, given that the aim of the present study was to compare AS-OCT and GS-1—both of which can be performed by orthoptists—we focused our analyses on these two modalities. We treated AS-OCT as a relative “true value” because prior reports have shown good sensitivity relative to manual gonioscopy and because GS-1 imaging is performed under bright conditions, which may underestimate angle closure. Nevertheless, since AS-OCT is not the true gold standard, we have revised and expanded the Discussion to acknowledge this limitation with the following statement:

“Comparisons with manual gonioscopy were not performed, and uncertainty remains regarding the extent to which the GS-1 captures the angle in a more open configuration compared with manual gonioscopy. In addition, because AS-OCT was used as a relative reference standard, verification bias is possible. Therefore, future studies are needed that include three-way comparisons among GS-1, manual gonioscopy, and AS-OCT.” (page 19, lines 296-300)

3. The manuscript is generally well-written, though minor grammatical refinements could help with better understanding.

Thank you for your helpful comments. We have carefully revised the manuscript for clarity and grammar.

We sincerely thank the academic editor, editors, and reviewers for their thorough and insightful review, which has greatly improved the quality of our paper. We remain open to any further revision requests and look forward to your response.

Sincerely,

Yuki Takagi, MD

Department of Ophthalmology

Japan Community Healthcare Organization Chukyo Hospital

1-1-10 Sanjo Minami-ku Nagoya-city, Aichi Prefecture, Japan

Telephone: +81-52-691-7151

Fax: +81-52-692-5220

Email: ytakagi@sanjogroup.jp

---

## [Editor Report · Decision Letter 2]

3 Sep 2025

Comparison of angle-closure detection between automated gonioscopy and anterior-segment optical coherence tomography

PONE-D-25-17859R2

Dear Dr. Takagi

We’re pleased to inform you that your manuscript has been judged scientifically suitable for publication and will be formally accepted for publication once it meets all outstanding technical requirements.

Kind regards,

Jiro Kogo

Academic Editor

PLOS ONE
---

## [Editor Report · Acceptance letter]

PONE-D-25-17859R2

PLOS ONE

Dear Dr. Takagi,

I'm pleased to inform you that your manuscript has been deemed suitable for publication in PLOS ONE. Congratulations! Your manuscript is now being handed over to our production team.

Kind regards,

on behalf of

Prof. Jiro Kogo

Academic Editor

PLOS ONE